# Natural-Target-Mimicking Translocation-Based Fluorescent Sensor for Detection of SARS-CoV-2 PLpro Protease Activity and Virus Infection in Living Cells

**DOI:** 10.3390/ijms25126635

**Published:** 2024-06-17

**Authors:** Elena L. Sokolinskaya, Olga N. Ivanova, Irina T. Fedyakina, Alexander V. Ivanov, Konstantin A. Lukyanov

**Affiliations:** 1Shemyakin-Ovchinnikov Institute of Bioorganic Chemistry, Miklukho-Maklaya 16/10, 117997 Moscow, Russia; elena.sokolinskaya@gmail.com; 2Engelhardt Institute of Molecular Biology, Russian Academy of Sciences, 119991 Moscow, Russia; olgaum@yandex.ru (O.N.I.); aivanov@yandex.ru (A.V.I.); 3Gamaleya National Research Centre for Epidemiology and Microbiology of the Ministry of Russia, 132098 Moscow, Russia; irfed2@mail.ru

**Keywords:** COVID-19, SARS-CoV-2, coronavirus, protease PLpro, live cell imaging, genetically encoded probes, green fluorescent protein

## Abstract

Papain-like protease PLpro, a domain within a large polyfunctional protein, nsp3, plays key roles in the life cycle of SARS-CoV-2, being responsible for the first events of cleavage of a polyprotein into individual proteins (nsp1–4) as well as for the suppression of cellular immunity. Here, we developed a new genetically encoded fluorescent sensor, named PLpro-ERNuc, for detection of PLpro activity in living cells using a translocation-based readout. The sensor was designed as follows. A fragment of nsp3 protein was used to direct the sensor on the cytoplasmic surface of the endoplasmic reticulum (ER) membrane, thus closely mimicking the natural target of PLpro. The fluorescent part included two bright fluorescent proteins—red mScarlet I and green mNeonGreen—separated by a linker with the PLpro cleavage site. A nuclear localization signal (NLS) was attached to ensure accumulation of mNeonGreen into the nucleus upon cleavage. We tested PLpro-ERNuc in a model of recombinant PLpro expressed in HeLa cells. The sensor demonstrated the expected cytoplasmic reticular network in the red and green channels in the absence of protease, and efficient translocation of the green signal into nuclei in the PLpro-expressing cells (14-fold increase in the nucleus/cytoplasm ratio). Then, we used PLpro-ERNuc in a model of Huh7.5 cells infected with the SARS-CoV-2 virus, where it showed robust ER-to-nucleus translocation of the green signal in the infected cells 24 h post infection. We believe that PLpro-ERNuc represents a useful tool for screening PLpro inhibitors as well as for monitoring virus spread in a culture.

## 1. Introduction

COVID-19 (coronavirus disease 2019) is an acute respiratory infection caused by the SARS-CoV-2 virus, which caused a pandemic in 2020 and led to enormous socio-economic consequences for humanity. Currently, the number of deaths since the beginning of the pandemic across the entire human population is more than 7 million people [1].

SARS-CoV-2 is an enveloped, single-stranded (+)RNA Betacoronavirus. To enter a host cell, the SARS-CoV-2 virion binds to the ACE2 receptor on the cell surface. After a 30 kb long RNA genome is released into the cell cytoplasm, the translation of a replicase gene begins. The replicase gene encodes two main open reading frames—ORF1a and ORF1b—which give rise to 16 non-structural proteins (nsp1–nsp16) necessary for viral RNA synthesis [2]. Cap-dependent translation results in one of two polyproteins—pp1a or pp1ab—which are further cleaved into nsp proteins by SARS-CoV-2 proteases. Coronavirus has two proteases—a 33-kDa chymotrypsin-like protease Mpro (the main protease), encoded by nsp5, and a 36-kDa papain-like protease (PLpro), which is the part of multi-domain, membrane-bound protein nsp3 [3]. PLpro cleaves the viral polyprotein precursors, pp1a and pp1ab, at three sites, which results in liberation of the first non-structural proteins (nsp1–nsp4) essential for the viral replication process [4], while Mpro cuts polyproteins at 11 sites, generating the remaining 12 nsps [3]. In addition to proteolytic activity, PLpro exhibits deubiquitinating and deISG15ylating (interferon-induced gene 15) activities that are responsible for host immune system suppression, facilitating viral proliferation and replication [5]. All of the above functions of PLpro, including the fact that it represents the earliest viral enzymatic activity which triggers all subsequent events, make PLpro an excellent target for anti-SARS-CoV-2 therapeutics.

The 250-amino-acid PLpro domain is a part of nsp3, the largest protein (1945 amino acid residues) encoded by the SARS-CoV-2 genome (Figure 1). Nsp3 acts as a membrane-anchored protein with four transmembrane segments that pass through the endoplasmic reticulum (ER) membrane, while most of the protein is exposed to the cell cytosol [3].

The importance of proteases has led to extensive efforts in the development of screening systems to study their enzymatic activity, as well as to test protease inhibitors as drug candidates. To study the activity of cellular proteases, genetically encoded fluorescent sensors are widely used [6]. The ability to use fluorescence microscopy in experiments with living cells makes it possible to study dynamic processes in real time. Due to the high pathogenicity of SARS-CoV-2, experiments with the live virus should be conducted at biosafety level 3 (BSL-3), which seriously limits the possibility of screening in the laboratory environment. In this case, screening systems based on genetically encoded fluorescent sensors in mammalian cells are highly effective, safe and almost the only tools for performing this task. Such screening systems have two main advantages. Firstly, live-cell-based experiments allow compounds that have a cytotoxic effect to be excluded, as well as compounds that are impermeable to the cell membrane. Secondly, the cleavage of the substrate in the cytoplasm of the cell is physiologically similar to the process of cleavage of the viral polyprotein by protease in a virus-infected cell [7].

Recently, FlipGFP-based screening systems have been developed to test inhibitors of coronavirus proteases PLpro [7] and Mpro [8]. The reporter system is based on the FlipGFP protein, which generates a fluorescent signal only after beta-strand reorientation [9]. The linker between the 10th and 11th beta-strands of the FlipGFP, crucial for reorientation, contains a PLpro or Mpro cleavage site. In the absence of active protease, no visible fluorescence was observed during sensor expression in mammalian cells. Protease expression led to β10-11 reorientation with formation of a functional GFP β-barrel, resulting in a noticeable green fluorescent signal in the system. When inhibitors were added, a dose-dependent decrease in fluorescence was observed. This system allows the screening of antiviral chemicals at biosafety level 2 (BSL-2) [8,9].

FRET sensors are also used as tools for monitoring protease activity in real time. The assay is typically based on a FRET pair of fluorescent proteins separated by a protease cleavage site, the proteolysis of which results in the disappearance of FRET. A FRET pair of cyan and yellow fluorescent proteins (CFP-YFP) separated by the Mpro recognition site was used to develop a reporter for the main protease activity [10]. When the FRET sensor was expressed in mammalian cells, a yellow fluorescent YFP signal generated as a result of the energy transfer from the excited CFP donor to the fused YFP acceptor was detected in the system. Separation of the two fluorescent proteins in the case of linker proteolysis by Mpro led to FRET signal disruption. The addition of an Mpro inhibitor (GC376) blocked the cleavage of the CFP-YFP linker and thus maintained the FRET signal in the system [10]. Thus, the following FRET-based assay can be considered as a platform for in vitro compound testing and the primary screening of potential Mpro inhibitors.

Intracellular translocation-based sensors provide another convenient way to detect protease activity in live cells. Many of them are based on a fluorescent protein with NLS that is retained in cytoplasm but translocates into the nucleus upon cleavage by viral protease. The cleavage sites correspond to polypeptide sequences in a polyprotein of the respective virus or to proteins that are targeted by viral protease. One of the pioneer studies in this field developed a sensor based on an antiviral protein, MAVS (also known as IPS-1, Cardif or VISA), which is localized on the outer mitochondrial membrane and interacts with cytosolic RNA helicase RIG-I, which senses double-stranded viral RNA [11]. As naive MAVS is targeted by the NS3 protease of the hepatitis C virus, upon HCV infection, the fused EGFP-MAVS sensor showed redistribution of green fluorescence from mitochondrial networks to diffuse cytoplasm staining, while RFP-NLS-MAVS allowed the monitoring of the translocation of RFP to nuclei shortly within 10–16 h post infection. Recently, a similar sensor was developed for the detection of enterovirus 71 infection, whose 2A protease also cleaves MAVS [12]. This sensor could potentially be used for a plethora of viruses as MAVS was shown to be digested by proteases of cytomegalovirus [13], MERS [14], hepatitis A virus [15,16] and other infections [17]. Notably, similar sensors with other proteins of the antiviral response of a host cell targeted by infections were also described [18].

Notably, the SARS-CoV-2 PLpro domain, as well as its viral protein targets, is situated near the cytoplasmic surface of the ER (Figure 1). Such a compartmentalization can be functionally important, bringing all the players into the same space with the 2D-like characteristics of diffusion of the membrane-anchored proteins. We recently developed a translocation-based sensor of PLpro activity, which was attached to the ER membrane by the tail-anchor (TA) short sequence [19]. Here, we design an improved version of this sensor that closely recapitulates the native substrate of PLpro and ensures high-contrast detection of the PLpro-triggered ER-to-nucleus translocation of the fluorescent signal in live cells. 

## 2. Results

### 2.1. Sensor Design

Our previously developed translocation-based sensor for PLpro activity [19] consisted of a fusion of the green mNeonGreen and red mScarlet I fluorescent proteins, which are characterized by fast maturation and high brightness in mammalian cells [13,14]. The PLpro recognition site was inserted between mNeonGreen and mScarlet I, whereas the whole sensor was attached to the cytoplasmic surface of the ER by a short membrane localization signal (the so-called “tail anchor”, TA) from the tyrosine phosphatase PTP1B. As a result, the sensor localized to the ER, and cleavage by PLpro liberated mNeonGreen to be diffusely distributed throughout the cell, including the nucleus. Thus, nucleus-to-cytoplasm ratios in green and red channels can be used as a measure of PLpro activation [19]. 

Here, we present a new sensor variant called PLpro-ERNuc, which was improved in two ways. First, we used a natural-target-mimicking attachment to the ER to ensure efficient cleavage by the native viral PLpro. The sensor design was changed by replacing the TA sequence with a fragment of the original sequence from the SARS-CoV-2 nsp3 molecule (amino acid residues 1341–1567) (Figure 2A). mNeonGreen and mScarlet I fluorescent proteins were separated by the linker containing a PLpro cleavage site with an LKGGAPTKV motif that corresponds to the nsp2–nsp3 junction from the SARS-CoV-2 polyprotein. Second, we added a nuclear localization signal (NLS) to the C-terminus of the reporter, which ensures accumulation of mNeonGreen in the nucleus after cleavage by PLpro. Thus, a clearer cytoplasm-to-nucleus translocation is expected for PLpro-ERNuc.

### 2.2. Testing PLpro-ERNuc Sensor in Cells Expressing Recombinant PLpro

The first series of experiments with the sensor was performed on mammalian cells transfected with PLpro. To mimic the natural localization of the viral PLpro, we generated the PLpro-Plus genetic construct containing the PLpro catalytic domain and adjacent nsp3 domains including ER transmembrane regions TM1-4 (Figure 2B). The blue fluorescent protein mTagBFP2 [20] separated by the “self-cleaving” T2A peptide [21] was used for the selection of transfected mammalian cells. Co-expression of PLpro-Plus and PLpro-ERNuc was expected to result in proximity of the protease domain to the target cleavage site (Figure 2C).

To prove the ER localization of PLpro-ERNuc, we performed cotransfection of the sensor with an ER marker. For this purpose, we used moxBFP protein [22], a monomeric blue fluorescent protein optimized for oxidizing environments, which was directed to the ER lumen with the prolactin signal sequence at the N-terminus and ER retrieval motif KDEL at the C-terminus. As expected, we observed good colocalization (Pearson’s coefficient 0.93) of PLpro-ERNuc (green and red channels) and moxBFP (blue channel) (Figure 3) (note that the resolution of regular fluorescence microscopy is not sufficient to distinguish between probes localized in the ER lumen and on the ER surface). 

To test the protease response, we generated a stable HeLa cell line expressing PLpro-ERNuc using the lentiviral transduction method. This line was then transiently transfected with the PLpro-Plus vector. The next day, the PLpro-Plus expression was checked by the presence of the blue fluorescent signal, whereas the sensor signal was assessed in two channels: red (mScarlet I) and green (mNeonGreen). 

In control cells expressing PLpro-ERNuc only, an expected cytoplasmic reticular network (ER) fluorescent signal was observed in both red and green channels (Figure 4A). In cells which also expressed SARS-CoV-2 PLpro protease, the red fluorescence was still localized to the ER. At the same time, the green signal was no longer associated with the reticular network but became fully accumulated in the cell nucleus (Figure 4A). To quantify the observed translocation, we calculated the nucleus/cytosol ratio in the green channel and normalized it to the nucleus/cytosol ratio in the red channel for the same regions of interest (ROIs) as previously suggested [19] (Figure 4B). The calculated ratio for PLpro-ERNuc increased about 14.3 times in the cells with PLpro expression compared to the rested cells (mean ratio before translocation—1.20, after translocation—17.17). 

We concluded that the observed changes fully corresponded to the expected PLpro-catalyzed cleavage of the sensor into the nuclear-localized mNeonGreen protein and the ER-bound nsp3-mScarlet I part. Ratio calculations confirmed a robust high-contrast readout provided by the reporter.

### 2.3. Testing PLpro-ERNuc Sensor in the Model of SARS-CoV-2 Infection of Cultured Cells

Next, we evaluated the performance of the reporter in the context of SARS-CoV-2 infection in the human hepatoma Huh7.5 cell line, which is permissive to SARS-CoV-2 infection [23]. To increase the reproducibility of imaging experiments, we generated a stable line of Huh7.5 cells expressing PLpro-ERNuc by lentiviral transduction. The cells were infected with SARS-CoV-2 virus that was removed two hours post infection (h.p.i.). Unlike the previous series of experiments where the proteolytic activity was provided by a genetically encoded protease (PLpro-Plus construct), here, the linker cleavage in the sensor was performed by viral protease expressed during SARS-CoV-2 replication. Confocal fluorescent imaging demonstrated that, in mock-treated cells both red and green fluorescent signals were localized in the ER network (Figure 5). Imaging at 24 h.p.i. showed that, in some cells, the green reticular network mostly disappeared, and the green fluorescent signal accumulated in the nucleus (Figure 5). Notably, in the SARS-VoV-2-infected cells treated with the antiviral agent molnupiravir [24], no significant translocation of the sensor occurred (Figure 5). 

The obtained results are consistent with the above genetically encoded protease experiments and confirm the effectiveness of the sensor in experiments with the active virus.

## 3. Discussion

Fully genetically encoded fluorescent sensors represent powerful tools that enable direct detection of various activities in living cells without the addition of any external chemicals or antibodies. Such sensors for viral proteases can be used in two main areas: (i) high-throughput or high-content screening of protease inhibitors (antivirals) and (ii) monitoring the spread of viral infection at the single-cell level (mostly in cell culture models). Genetically encoded sensors of protease activities are commonly based on the three main fluorescence readouts, namely, the disappearance of FRET between two fluorescent proteins, changes of the intracellular localization (translocation) of a fluorescent protein and fluorescence turn-on due to fluorescent protein polypeptide chain rearrangement and chromophore maturation (FlipGFP). Each approach has advantages and shortcomings. At the molecular level, only FRET sensors ensure an immediate response to the protease cleavage, whereas intracellular translocation occurs within minutes, and FlipGFP maturation requires tens of minutes. At the same time, FRET sensors possess the lowest dynamic range (usually 1.5–1.8-fold changes), and much more contrasting and reliably detectable changes are characteristic for translocation sensors and FlipGFP. The main advantage of FlipGFP-based sensors is a strong (~70-fold) increase in fluorescence upon protease cleavage, which makes it suitable for high-throughput screening using fluorescence plate readers and FACS. Translocation-based sensors are most flexible in terms of a choice of the colors since regular fluorescent proteins of any desirable colors and their combinations can be used.

Here, we developed a translocation-based fluorescent sensor, PLpro-ERNuc, specifically designed for the SARS-CoV-2 PLpro protease. As the PLpro-ERNuc design implies ER-anchored localization, we did not perform experiments on characterization of the bacteria-expressed and purified sensor. However, we observed no cleavage and translocation of the sensor in the absence of PLpro in either HeLa or Huh7.5 cells. Thus, the LKGGAPTKV motif in the linker is not recognized by endogenous cellular proteases, which ensures reporter specificity. Compared to previously published PLpro sensors, PLpro-ERNuc has several advantages. Firstly, the new sensor design implies an intracellular localization and protein context that are very close to that of the natural target of PLpro. Secondly, PLpro-ERNuc is characterized by a much higher dynamic range than our previous version of the sensor [19] (14- vs. 2-fold). In our experiments, individual cells showed varying degrees of ER-to-nucleus translocation of mNeonGreen (see the widely distributed ratio values in Figure 4B). This heterogeneity likely resulted from the transient transfection procedure with the PLpro-Plus plasmid. Indeed, the cell begins to express it only after cell division, when the plasmid enters the nucleus. Thus, cells showing the lowest ratio values may have initiated PLpro-Plus expression shortly before imaging. In Figure 4 and Figure 5, one can notice a considerable accumulation of mNeonGreen-NLS signal in nucleoli. This phenomenon has been reported for some fluorescent proteins fused with an NLS or a nuclear protein due to their dynamic electrostatic interactions with RNA [25,26]. To simplify analysis, we used the ROI over the entire nucleus including both relatively dim nucleoplasm and bright nucleoli. Alternatively, ROIs for either nucleoplasm or nucleoli only can be applied with similar results. Confocal microscopy (see Figure 5) provides a clearer picture of the probe distribution between nucleus and cytoplasm due to suppression of out-of-focus fluorescence, and, thus, the lower background level of nuclear fluorescence in the resting cells. However, we showed that wide-field microscopy is also applicable for assessing the sensor (see Figure 4). Therefore, scientists with access to different microscopes can use PLpro-ERNuc.

PLpro-ERNuc demonstrated a robust readout in both overexpressed protease and SARS-CoV-2 infection models. As PLpro-ERNuc-based screening can be performed in living cells, it is potentially suitable for simultaneously assessing the cytotoxicity and cell membrane permeability of the tested compounds. Sensors for visualization of infected cells are highly needed by virologists as they allow the monitoring of virus spread in a culture. First, it allows protease activity to be measured in the context of expression with other viral proteins and the virus-altered intracellular membrane milieu, which could affect localization of the protease and its activity. Second, this can show the kinetics of virus spread in cell culture. Moreover, for some infections, it may show direct cell-to-cell transmission through tight junctions [27], leading to the identification of yet unknown mechanisms of transmission and investigation of mechanisms of antiviral drug resistance. In addition, screening platforms based on native virus application possess several benefits. During a coronaviral infection, PLpro is expressed in the cell along with many other viral proteins. It means that cross-viral protein interactions can affect the potential protease inhibitor. Additionally, coronaviral infection induces significant cellular membrane rearrangements that do not happen during the transient transfection of a protease-encoding plasmid.

## 4. Materials and Methods

### 4.1. General Methods and Materials

DNA oligonucleotides for the PCR were commercially synthesized by Evrogen (Moscow, Russia). An Encyclo PlusPCR kit (Evrogen) was used for the routine PCR amplifications. ScreenMix-HS with HS Taq (Evrogen) was used for the screening of *E. coli* colonies after the transformation. The length of the PCR products was verified on 1.2% TAE agarose gels. The PCR products of interest were then cut and purified with a Cleanup Mini kit (Evrogen). The DNA plasmids were cultured overnight and then extracted using a Plasmid Miniprep kit (Evrogen). The gene sequences were verified by Sanger sequencing. Huh7.5 cells were a kind gift from C. M. Rice (Rockefeller University, New York, NY, USA) and Apath L.L.C. (St. Louis, MO, USA). SARS-CoV-2 (passage 4) corresponding to the hCoV-19/Russia/Moscow-PMVL-12/2020 strain with infectivity of 10^6^ TCID50/mL was from the Russian State collection of viruses at the National Research Center for Epidemiology and Microbiology. 

### 4.2. DNA Cloning

The DNA constructs were cloned using the Golden Gate cloning system [28]. The genes of interest were cloned into Level 0 and Level 1 backbones obtained from a MoClo Toolkit (Kit #1000000044 from AddGene (Watertown, MA, USA)). BpiI (BbsI) and Eco31I (BsaI) restriction endonucleases (Thermo Scientific, Waltham, MA, USA) and T4 DNA ligase (Evrogen) were used for the MoClo procedure.

The expression vector encoding SARS-CoV-2 nsp3 was a gift from Fritz Roth (AddGene plasmid #141257 [29]). PLpro-coding DNA fragments were generated by PCR amplification on the template of the nsp3 coding sequence and included ubiquitin-like domain 2 (Ubl2, residues 746–805), PLpro catalytic domain (residues 806–1058), nucleic-acid-binding domain (NBD, residues 1059–1200), marker domain (MD, residues 1201–1340) and four transmembrane regions (TM1-4, residues 1341–1567) [30]. Transfection efficiency was visualized using mTagBFP2 protein fused to the “self-cleaving” T2A peptide GSGEGRGSLLTCGDVEENPGP [21].

DNA constructs for the PLpro-ERNuc consisted of two fluorescent proteins (mNeonGreen and mScarlet I) connected by an amino acid linker GGGST-LKGGAPTKV-GGSGS, where LKGGARTKV corresponded with the nsp2–nsp3 junction from the SARS-CoV-2 polyprotein. On the N-terminus, the sensor contained four transmembrane regions (TM1-4, residues 1341–1567) from SARS-CoV-2 nsp3, which ensured ER localization of the sensor, and three Y-domains (Y1-3, residues 1568–1945), which provided natural context for the nsp2–nsp3 cleavage site. On the C-terminus of the sensor, the 26-amino-acid nuclear localization signal (NLS) was added (DPKKKRKVDPKKKRKVDPKKKRKSGL), which provided green signal accumulation in the nucleus after proteolysis.

All elements of the translation unit including coding sequences, promoter and terminator were assembled into Level 1 expression vectors using the Golden Gate cloning system. A DNA construct encoding the PLpro was put under the control of a CMV promoter and possessed a SV40 poly(A) sequence. The DNA construct encoding the PLpro-ERNuc was put under a CMV promoter and was assembled into a pLVT-like vector for further lentiviral production.

### 4.3. Cell Culture and Transfection

Imaging experiments reported in the manuscript were performed with HeLa and Huh-7.5 cells. HEK293T cells were used for lentivirus production. HeLa and HEK293T cells were cultured in complete DMEM (Dulbecco’s Modified Eagle Medium) (PanEco, Moscow, Russia) supplemented with 10% fetal bovine serum (FBS) (BioSera, Nuaille, France), 100 U/mL penicillin and 100 mg/mL streptomycin (PanEco) at 37 °C in 5% CO_2_. Huh-7.5 cells were cultured in Dulbecco’s Modified Eagle Medium (PanEco) supplemented with 10% FBS (BioSera), 2 mM L-Glutamine (Gibco, Waltham, MA, USA), 100 U/mL penicillin and 100 mg/mL streptomycin (PanEco) at 37 °C in 5% CO_2_. For the live cell imaging experiments, the DMEM was replaced by imaging media: MEM (PanEco) supplemented with 10% FBS (BioSera) and 20 mM HEPES (Corning, NY, USA). Transfections were performed in Opti-MEM (Gibco) with GenJect-39 (Molecta, Moscow, Russia) transfection reagent according to the manufacturers’ protocol (1:1 DNA/GenJect-39 ratio).

### 4.4. Lentivirus Production and Stable HeLa Cell Line Generation

Lentivirus was produced from a lentiviral pLVT-like vector transfected into HEK293T cells using a polyethyenimine (PEI) transfection protocol. HEK293T cells were seeded at 3.8 × 10^6^ cells per plate in 10 cm tissue culture plates (SPL Life Sciences, Gyeonggi-do, Korea) in complete DMEM at 37 °C, 5% CO_2_. After 24 h, cells were transfected in Opti-MEM (Gibco) with a mixture of 3 transfection plasmids (pR8.91, pMDG, lentiviral transfer plasmid encoding PLpro-ERNuc) with PEI 25K (Polysciences, Warrington, PA, USA) according to the manufacturers’ protocol (1:3 DNA/PEI ratio). The virus was harvested at 48 h post transfection and was concentrated by ultracentrifugation at 100,000× *g* (Beckman, Brea, CA, USA) at 4 °C for 3 h. 

For lentiviral transduction, HeLa cells were incubated with the lentivirus for 24 h. Fluorescence-activated cell sorting (FACS) was performed using BD FACSAria III (BD Biosciences, Franklin Lakes, NJ, USA).

### 4.5. SARS-CoV-2 Infection Assay

Huh7.5 cells were seeded on 10 cm dishes (SPL Life Sciences, Gyeonggi-do, Republic of Korea) at a density of 1 × 10^6^ in DMEM supplemented with 10% FBS. When the cells reached 50% confluence, they were transduced with the lentivirus. Three days post transduction, the cells were harvested and seeded in glass-bottom cell culture dishes with 15 mm inserts (Nest, Wuxi, China) in the same medium.

Infection with SARS-CoV-2 was performed as described in [23]. Briefly, when the cells on the confocal dishes reached 40% density, the virus was added for two hours in DMEM with 2.5% serum at a multiplicity of infection (MOI) of 0.1. For molnupiravir treatment, the cells were infected with SARS-CoV-2, and 5 µM molnupiravir (Sigma-Aldrich, Burlington, MA, USA) was added 4 h.p.i., when the medium with the virus was replaced with fresh medium. Fluorescence was visualized 24 h.p.i. The cells were fixed with methanol/acetone, as described in [31], followed by transfer to PBS. Confocal 8-bit digital images were acquired using a Leica TCS SP5 confocal laser-scanning microscope (Leica Microsystems, Wetzlar, Germany) equipped with an HCX PLAPO CS 63× 1.4NA oil immersion lens. Fluorescence was excited at 488 nm and registered in the 510–535 nm range in the green channel and excited at 543 nm and registered in the 570–600 nm range in the red channel.

### 4.6. Live Cell Imaging

For the live cell imaging experiments, an HeLa stable line expressing the PLpro-ERNuc sensor was used. Cells were seeded on glass-bottomed 35 × 10 mm dishes (SPL Life Sciences) and were incubated at 37 °C in 5% CO_2_ overnight. The next day, cells were transfected with the PLpro-Plus plasmid. Imaging experiments were performed 24 h post transfection using a BZ-9000 inverted fluorescence microscope (Keyence, Osaka, Japan). The imaging was carried out at 37 °C with a 60× PlanApo 1.40NA oil objective (Nikon, Melville, NY, USA). A TexasRed OP79302 SB filter (excitation 562/40 nm, emission 624/40), GFP-BP OP66836 BZ filter (excitation 470/40 nm, emission 535/50) and 49021-ET-EBFP2/Coumarin/Attenuated DAPI filter (excitation 405/10 nm, emission 460/25) (Chroma Technology, Bellows Falls, VT, USA) were used to visualize mScarlet I, mNeonGreen and mTagBFP2, respectively.

### 4.7. Image Analysis

The images were processed using Fiji ImageJ version 2.14.0/1.54f [32]. For the calculation of the nucleus/cytoplasm ratio, the mean fluorescence was measured in the nucleus and cytoplasmic ROIs, respectively. The same ROIs in the green and red channels were used for each individual cell. For all calculations, the mean fluorescence intensity of the background was subtracted out of the cells. For experiments stating *p*-values, a paired Student’s *t*-test was performed. *p*-values < 0.05 were considered as significant.

## 5. Conclusions

Here, we used a bioinspired design to construct PLpro-ERNuc, a new genetically encoded fluorescent sensor for the monitoring of the live cell activity of the SARS-CoV-2 PLpro protease. A fragment of nsp3 protein including four ER-associated transmembrane regions was used to place the sensor at an intracellular localization and protein context highly similar to that of the native target of PLpro. The use of two fluorescent proteins, one which remains attached to ER all the time and another which translocates from the ER to the cell nucleus upon sensor cleavage with PLpro, enabled an easy-to-detect, high-contrasting ratiometric (and thus independent on the sensor expression level) readout. We demonstrated the efficient functioning of PLpro-ERNuc in both recombinant PLpro overexpression and SARS-CoV-2 virus infection cell models. These two models correspond to the main areas of viral protease sensor application, namely, safe, virus-free, cell-based screening of protease inhibitors to develop novel antivirals and monitoring virus spreading for basic studies. 

## Figures and Tables

**Figure 1 ijms-25-06635-f001:**
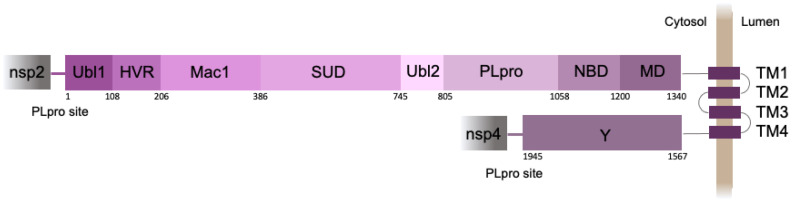
Schematic representation of SARS-CoV-2 nsp3 domain organization. Ubl1—ubiquitin-like domain 1, HVR—hypervariable region, Mac1—macrodomain 1, SUD—SARS-unique domain, Ubl2—ubiquitin-like domain 2, PLpro—papain-like protease, NBD—nucleic-acid-binding domain, MD—marker domain, TM 1–4—transmembrane regions 1–4, Y—the Y domain. Numbers indicate the amino acid positions; domains are colored in different shades of purple.

**Figure 2 ijms-25-06635-f002:**
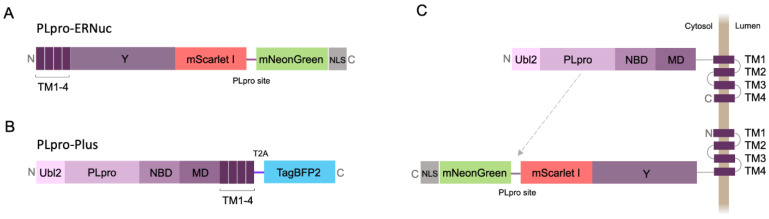
Diagrams of the genetic constructs. PLpro-ERNuc (**A**) and PLpro-Plus (**B**) constructs used for imaging experiments in mammalian cells and their proposed intracellular localization (**C**). Indication of nsp3 domains is described in Figure 1. mScarlet I, mNeonGreen, mTagBFP2—red, green and blue fluorescent proteins, respectively, shown by rectangle of corresponding colors. T2A—“self-cleaving” T2A peptide, NLS—nuclear localization signal. The gray arrow indicates targeting of the PLpro cleavage site by the protease.

**Figure 3 ijms-25-06635-f003:**
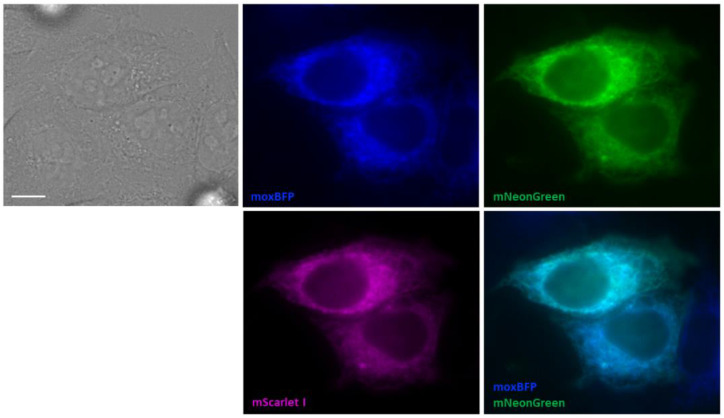
Colocalization of PLpro-ERNuc with the ER marker moxBFP. HeLa cells were cotransfected with PLpro-ERNuc and moxBFP plasmids. Fluorescent images in blue, green and red channels are shown together with overlay of the blue and green channels (Pearson’s correlation coefficient 0.93). Scale bars, 10 μm.

**Figure 4 ijms-25-06635-f004:**
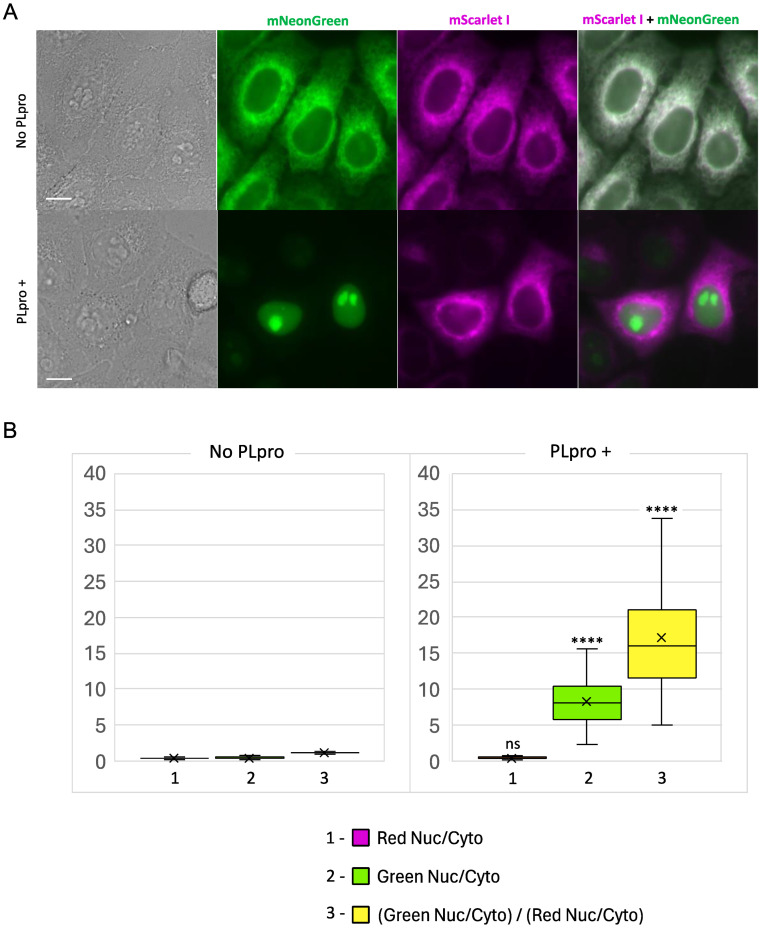
PLpro-ERNuc translocation-based fluorescent sensor for SARS-CoV-2 PLpro protease activity. (**A**) Representative wide-field fluorescence cell images of HeLa cells expressing PLpro-ERNuc. The upper panel: HeLa cells expressing PLpro-ERNuc in the absence of PLpro protease. The lower panel: HeLa cells expressing PLpro-ERNuc transiently transfected with PLpro-Plus. Scale bars, 10 μm. (**B**) Quantification of nucleus/cytosol ratios for cells expressing PLpro-ERNuc in the absence of PLpro (left, n = 107 cells) and in the presence of PLpro (right, n = 103 cells). In the box plot, median (horizontal line inside each box), mean (cross inside each box), 1st and 3rd quartiles as well as minimum and maximum values are shown. The purple (1) and green (2) boxes are the ratio of fluorescence in the nucleus to the cytoplasm in the red and green channels, respectively, and the yellow (3) boxes are the ratio of the obtained values to each other. Statistical significance is shown in the right graph in comparison with corresponding values in the left graph (****—*p*-value < 0.0001, ns—not significant).

**Figure 5 ijms-25-06635-f005:**
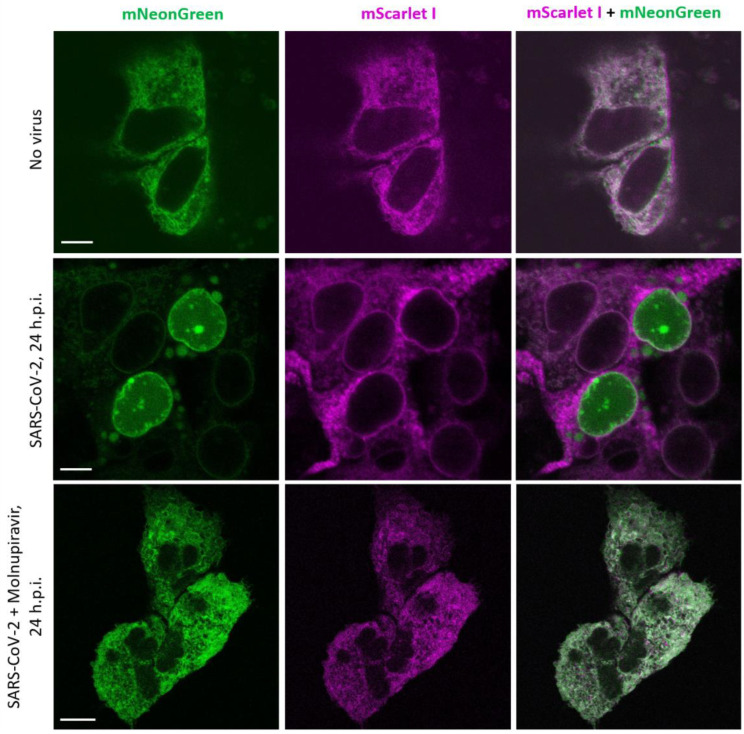
Assessment of SARS-CoV-2 infection of Huh7.5 cells expressing PLpro-ERNuc. Confocal microscopy of fixed cells in the green and red channels in the absence of SARS-CoV-2 infection (upper panels); SARS-CoV-2-infected cells, 24 h.p.i. (middle panels); and SARS-CoV-2-infected and molnupiravir-treated cells, 24 h.p.i. (bottom panels). Four independent experiments on virus infection were performed (3 Petri dishes and about 50 cells analyzed in each); representative cells are shown. Scale bars, 10 μm.

## Data Availability

Data is contained within the article.

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
