# Peer review of "Natural-Target-Mimicking Translocation-Based Fluorescent Sensor for Detection of SARS-CoV-2 PLpro Protease Activity and Virus Infection in Living Cells"

_ijms, 2024, doi:10.3390/ijms25126635_

Round 1

Reviewer 1 Report

Comments and Suggestions for Authors

These authors established a new fluorescent probe to detect SARS-CoV-2 PLpro protease. This probe has been improved due to the high anchoring efficiency in the ER membrane using the TA transmembrane domain compared to the previous one. The manuscript includes some concerns as follows:

1.       In the introduction, the authors mention the probe using GFP beta-barrel reconstruction or FRET, and the previous probe [citation #16], but in the discussion, they only discuss the increased dynamic range in light of the results in #16. This comparison with other probes should be made more explicit in the discussion. Moreover, how did the authors get the value (about 10-fold in L. 226); this should be explained in the discussion.

2.       In Fig. 3, the NLS-tagged GFP was localized not only in the nucleoplasm but also nucleous, and its fluorescent intensity in the nucleous was extremely high. This affects the intensity quantification so the authors should mention this caution, in addition to the reason why this protein was efficiently localized in the nucleous not only in the nucleoplasm.

3.       The microscopy images in Fig. 3 could be acquired using confocal or not wide-field ones.

4.       In Fig. 3 (yellow box), the author should discuss what was going on in the cells that showed the lowest values among the widely distributed ratio values.

5.       In Fig. 3, the description of the box plot should be described more clearly (for example, ‘X’ and the line).

6.       Fig. 4: Did the authors check the reproducibility of this experiment? It should be shown appropriately.

7.       Did the authors check the protease specificity of this probe in vitro (e.g., using purified proteins)?

Minors:

1.       This manuscript includes the Russian language in some places.

2.       The pseudocolor for the microscopic images for partial color blindness should be used.

Comments on the Quality of English Language

This manuscript includes the Russian language in some places.

Reviewer 2 Report

Comments and Suggestions for Authors

This article reported the development of a new genetically encoded fluorescent sensor, named PLpro-ERNuc, for detecting PLpro activity in living cells using translocation-based readout. 

Several suggestions:

1.      Figure 2A, there is no signal peptide in the N-terminus of the pLpro-ERNuc. Is there any evidence (e.g., co-localization of this reporter with E.R. marker) or reference to support that this reporter protein is indeed ER-localized?

2. In Figure 3, the comparison is done between no PLpro and PLpro+. It is better to compare the PLpro wild-type and inactive mutant (e.g., C111S mutant).

3. In Figure 3A and Figure 4, the nuclear mNeonGreen proteins seem to be located in the nucleolus. If yes, please explain it since NLS but not the nucleolus-localization signal has been added.

4.      Sub-cellular localization of a specific protein is never 100% as expected. To prove this reporter system is a useful tool for screening anti-PLpro, it is better to have inhibitor(s) against PLpro activity to verify this reporter system, e.g., inhibitors mentioned in reference 7, or in ChemBioChem 2022, e202200327, or in NATURE COMMUNICATIONS | (2021) 12:743. If none of these inhibitors is available, commercially available anti-viral agents (e.g., remdesivir, molnupiravir, lopinavir/ritonavir, or others mentioned in Turk J Med Sci. 2021 Dec 17;51(SI-1):3372-3390.) could be tested in the model of SARS-CoV-2 infection (Figure 4).

Round 2

Reviewer 1 Report

Comments and Suggestions for Authors

3.       The microscopy images in Fig. 3 could be acquired using confocal or not wide-field ones.

Response:

Indeed, due to the strong suppression of out-of-focus fluorescence, confocal microscopy provides a clearer picture of the probe distribution between nucleus and cytoplasm due to the lower background level of nuclear fluorescence in the resting cells. However, we aim to show that both wide-field and confocal microscopy modes are applicable to assess the sensor. Thus, scientists having access to different microscopes can use PLpro-ERNuc. 

Response by referee

Regarding this rebuttal, if the authors consider so, this reason should be added in the main text because the reason why confocal microscopy was used in lines 171-172 was described but the reason why wide-field microscopy was employed seems to be not.

5.       In Fig. 3, the description of the box plot should be described more clearly (for example, ‘X’ and the line).

Response:

Sorry, we are not sure what is unclear in our description.

Response by referee

Is "x" median or mean? And is the line near the center of the box median or mean? I suggested that it would be better to define them clearly.

1.       This manuscript includes the Russian language in some places.

Response:

We could use commentaries in Russian during preparation of the manuscript but it was expected to be removed from the final version. Now we double checked the manuscript but did not find any words in Russian. If we overlooked something please ignore/delete it.

Response by referee

There is in Line 43 (ORF1a и ORF1b). It remains.

Reviewer 2 Report

Comments and Suggestions for Authors

This revised manuscript has addressed most of the issues I have raised previously except issue 2 [In Figure 3, the comparison is made between no PLpro and PLpro+. It is better to compare the PLpro wild-type and inactive mutant (e.g., C111S mutant).] due to the limited time for revision.

Author Response

We sincerely thank Reviewer for the criticism, which helped improve the manuscript!